# Development of a Novel Approach for Detection of Equine Lameness Based on Inertial Sensors: A Preliminary Study

**DOI:** 10.3390/s22187082

**Published:** 2022-09-19

**Authors:** Cristian Mihaita Crecan, Iancu Adrian Morar, Alexandru Florin Lupsan, Calin Cosmin Repciuc, Mirela Alexandra Rus, Cosmin Petru Pestean

**Affiliations:** 1Department of Surgery and Intensive Care, Faculty of Veterinary Medicine, University of Agricultural Sciences and Veterinary Medicine Cluj-Napoca, 400372 Cluj-Napoca, Romania; 2Department of Obstetrics and Reproduction, Faculty of Veterinary Medicine, University of Agricultural Sciences and Veterinary Medicine Cluj-Napoca, 400372 Cluj-Napoca, Romania

**Keywords:** equine lameness, lameness screening, lameness detector, inertial sensor, accelerometer, impulses

## Abstract

Both as an aid for less experienced clinicians and to enhance objectivity and sharp clinical skills in professionals, quantitative technologies currently bring the equine lameness diagnostic closer to evidence-based veterinary medicine. The present paper describes an original, inertial sensor-based wireless device system, the Lameness Detector 0.1, used in ten horses with different lameness degrees in one fore- or hind-leg. By recording the impulses on three axes of the incorporated accelerometer in each leg of the assessed horse, and then processing the data using custom-designed software, the device proved its usefulness in lameness identification and severity scoring. Mean impulse values on the horizontal axis calculated for five consecutive steps above 85, regardless of the leg, indicated the slightest subjectively recognizable lameness, increasing to 130 in severe gait impairment. The range recorded on the same axis (between 61.2 and 67.4) in the sound legs allowed a safe cut-off value of 80 impulses for diagnosing a painful limb. The significance of various comparisons and several correlations highlighted the potential of this simple, affordable, and easy-to-use lameness detector device for further standardization as an aid for veterinarians in diagnosing lameness in horses.

## 1. Introduction

Lameness, a clinical sign and not a disease per se [1], is among the most prevalent health problems affecting all horses, regardless of their age, breed, gender, or use [2]. In addition to its significant economic impact on owners and the whole equestrian industry, lameness represents a welfare problem for the animal, due to prolonged pain, and with possibly serious consequences on the horse’s overall health, especially when it becomes chronic [3], leading to the need for early diagnosis. Although the classical lameness examination is based on observation, with certain repeatability and reliability, clinicians are not able to capture small changes in locomotion patterns because of the relatively low image capturing frequency of the human eye [4]. As a materialization of the attempts to avoid the subjectiveness of human observation during classical equine lameness assessment [5] and to evaluate equine gait objectively by obtaining precise quantitative results for the investigated parameters, the use of several device systems, such as force plates, optical motion capture systems, and inertial measurement units have been described [6,7,8] in the scientific literature. Stationary force plates are often recommended as the most precise system, and are the gold standard for objective lameness evaluation in horses, providing accurate and reliable results with high sensitivity and specificity [9,10]. However, the costs and complexity of the apparatus need controlled conditions and multiple environmental restrictions, and the necessity for each limb to be sampled and assessed for complete examination separately poses major obstacles for the conventional veterinary practice in acquiring and using these plates. 

Optical motion capture systems represent another method to collect real-time gait information. However, the parameters’ precision, repeatability, and validity differ between analyses, reducing the accuracy and relevance of the data [11]. These systems are also expensive and can mostly be used in laboratory conditions.

In parallel with the technological innovation in microelectronics and wireless systems, inertial sensors, including accelerometers and gyroscopes, have received significant attention due to their versatile applicability in sport activities to measure the performance of athletes and other sports instruments [12,13,14,15,16], gait analysis [17,18], navigation techniques [19], aerospace industry, and medical devices [20,21]. The inertial sensor-based methods have been proven to be reliable in equine clinical practices, being relatively simple, easy to apply, non-invasive, and allowing for real-time data collection in a short period [22,23,24]. Thus, these sensors have the potential to become important supporting tools for both experienced and unexperienced equine veterinarians during the clinical examination of their patients [25,26,27,28], especially in such instances and situations in which the gait evaluation on force plates is not practical [28,29]. The sensor-based systems have to be coupled with software that is responsible for data processing and analysis.

In order to avoid interferences or perturbances of limb movements, the inertial sensor-based devices have to be as small and light as possible. The best approach we found was to mount the components together in a way that resembles the protective leg equipment used in sport horses, with which they are familiar. In addition, the signal transmission range should allow for the evaluation of the horse avoiding the need for cables, which would impose movement restrictions. The data processed by the sensor should be more sensitive than those that can be observed via clinical examination. The data collection, transmission, analysis, and representation should be fast and reliable to accurately quantify lameness.

Numerous devices based on inertial sensors are currently available on the market. Depending on their construction principles and brand, these differ in complexity, tracked kinematic parameters, positioning, and orientation of the horse’s body. However, the available literature data indicate that these devices are precise and reliable in detecting gait asymmetries indicative of several medical abnormalities [24,25,26,27,28,29,30].

The present study aimed to design, develop, and test an original device system, as a diagnostic aid to be used in field conditions, to record and interpret the modifications of locomotory mechanics in horses displaying lameness.

The analysis of the collected data showed that the mean number of impulses recorded on the X axis of the incorporated accelerometer was the most relevant, both in identifying lameness per se and in diagnosing its severity. As these values proved in all the assessed horses the presence of gait abnormalities indicative of lameness and did not miss the mildest lameness either (scored subjectively as 1/5 on the AAEP lameness scale), we consider that the Lameness Detector 0.1 has a good potential to be further improved and standardized to aid the veterinary lameness diagnosis in horses.

## 2. Materials and Methods

### 2.1. The Lameness Detector 0.1 Device System

The Lameness Detector 0.1 system was composed of four identical devices to be attached by adjustable and very flexible straps on the dorsal aspect of the pasterns of each assessed horse, to record and transmit certain characteristics of their gait to computer software to receive the data and allow its processing for interpretation. To interfere as little as possible with the natural gait, light and small components were interconnected for constructing the devices: a rechargeable battery (accumulator), an accelerometer, a microcontroller board, and a Bluetooth device. The custom-written computer software installed on a laptop was named Lameness Detector 0.1 software (open-source).

To protect the device from possible shocks and to keep the assembly together, the components were enclosed in a foldable case (appropriate boxes held together by hinges), as shown in Figure 1, made of plastic filament (polylactic acid, Plastic 2 Print) extruded through a nozzle which melted it, while being gradually deposited in a structured way on the build platform of a 3D printer (3D Printizer, Uzina3D Machines SRL). The OpenSCAD software (openscad.org) was used for designing the cases, allowing precise digital calculations and dimensioning of the plastic boxes. The case provided enough space for the cables too, and its upper end was left open to connect the lameness detector’s components by these cables. To prevent the movements of the accelerometers in their cases, which could have introduced errors in the measurements, these had been glued inside the custom-tight-made cases (using Loctite^®^ Super Glue Ultra Gel Control™).

The largest piece of the device was a rechargeable battery (an accumulator) meant to provide the electric power needed by all other components (Figure 1). Weighing only 27 g, the lithium-ion polymer rechargeable battery (LIPO 3.7 V 1400 mA, Olimex^®^, Plovdiv, Bulgaria) had dimensions of 50 × 34 × 8 mm, a 700 mA maximum charging current, a 1400 mA maximum discharge current (continuous), and needed a limited charge voltage of 4.2 V. This rechargeable battery was considered ideal for the Arduino board which has a factory built-in dedicated connector and included charger for this type of rechargeable battery.

The ADXL345 (Analog Devices, Inc.^®^, Wilmington, MA, USA) is a small (3 × 5 × 1 mm, weighting 20 g), ultra-low power (25 to 130 μA at Vs = 2.5 V) three-axis accelerometer (Figure 1), with high resolution (13-bit) measurement up to ±16 g, which senses both static and dynamic acceleration, and is thus usable as a tilt sensor (with a resolution of inclination changes down to a minimum of 0.25°) or to detect free fall. An important feature of this accelerometer for its use in the Lameness Detector 0.1 was that it can detect both the presence and the lack of motion by comparing acceleration values to user-defined thresholds. The sensor consumed 0.4 mA when in use (data transfer) and 0.25 mA when on standby, respectively.

The Arduino Pro Mini (SparkFun Electronics^®^, Boulder, CO, USA) microcontroller board (Figure 1) was chosen for its small dimensions (approximately 17 × 33 mm, weighing 2 g), low electric power needs (3.3 V and 8 MHz versions), and versatility. This board had a central role in the Lameness Detector 0.1, and it was connected to all the other components: power intake and distribution to the accumulator (FTDI^®^, Glasgow, UK, cable), data (impulses) collection (input) to the accelerometer, and data transmission (output) by wireless communication between the device and a computer to a Bluetooth gadget (Bluetooth Mate Gold for Arduino, class 1, SparkFun Electronics^®^) which had a transmission range of 100 m and constant power consumption of 25 mA. The lack of pre-mounted headers on the microcontroller board permitted the use of connectors to all the other components. Among the six analog and 14 digital pins (which can be used as chosen, as an input or output), the specialized functions of the pins 0 (RX) and 1 (TX) allowed the receival (RX) and transmission (TX) of transistor–transistor logic (TTL) serial data (from the accelerometer to the Bluetooth). These pins were connected to the TX-0 and RX-1 pins of the six-pin header. Arduino Pro Mini is an open-source hardware, with 32 kB of flash memory, of which 2 kB is used by the bootloader.

For this study, the board had been programmed with the Arduino Software (IDE, Arduino^®,^, version number 15, Monza, MB, Italy), which enables communication with a computer by serial communication or by the transmission of simple textual data using the included serial monitor. In this study, the software was set to collect data from the accelerometer at intervals of 10 ms, and to transmit the impulses to Bluetooth. The setting function of the software was only accessed at the board’s turn on or its reset, but throughout its functioning, the software ran in successive, continuous loops. During the software configuration, the following two sensor parameters were used for setting the communication with the accelerometer: writeTo (DATA_FORMAT, max_g) and writeTo (POWER_CTL, 0x08), as well as two instructions for the Bluetooth connection: Wire.begin (); Serial.begin (9600). During the board’s loop function, the reading of the sensor was programmed to be made by the readAccel command and the data were transfer by the Serial,write command. The data acquired from the sensor were processed to reduce the traffic dimension by the ‘do pack’ command inserted in the program.

The Lameness Detector 0.1 software was custom-designed and installed on a laptop to visualize the data recorded by the accelerometer, pre-processed by the microcontroller board, and sent by Bluetooth. The software’s main window was divided into four panels, one for each sensor (Figure 2). Each of the panels had three sub-panels to represent the graphically processed data recorded on the three axes (X, Y, and Z) of each accelerometer.

On each of the three axes, the acceleration values were shown in m/s^2^ and the trajectory of the sensor was displayed graphically (Figure 2). This representation allowed the comparisons between the lame and the sound contralateral limbs.

### 2.2. The Horses and Their Gait Assessment Protocol

All the procedures described in this study took place at a University Equine Hospital in the presence and with the informed consent of the owners who brought their horses for examination and treatment of lameness, with or without a previous diagnosis provided by a veterinary practician. Ten horses were selected for the study based on the following inclusion criteria: adult sport horses with previous training history that had been familiarized with the wear of protective leg equipment (bandages or Polo wrap-type protections, and boots such as brushing boots, tendon boots, or fetlock boots); displaying lameness in exclusively one leg (five horses with a front-leg lameness and five horses with a hind-leg lameness), and showing no sign of environment-related stress in the clinic or upon attaching the Lameness Detector 0.1 onto their legs.

The final study sample consisted of 6 mares and 4 stallions between 5 and 14 years of age, of which 4 were mixed breed sport horses, 3 Lipizzaners, 1 a Oldenburg, 1 a Holsteiner, and 1 a Romanian Sport Horse.

After a general clinical examination of the horses, the Lameness Detector 0.1 device system was fitted onto their legs (Figure 3). They were led to the lameness assessment area, a flat terrain covered with asphalt. The attachment system of the devices consisted of very flexible and adjustable straps which (because of the small dimensions and light weight of the devices) exerted only modest pressure on both the lame and healthy limbs (not more than the wear of protective hoof-boots, with which all horses had been previously familiarized). Thus, the influence on the horses’ locomotion and the effect of skin displacement on the recorded values were limited as much as possible.

The three axes of the accelerometers within the Lameness Detector 0.1 measured the forward–backward movement on the X axis, the upward–downward movement on the Y axis, and the abduction–adduction movement on the Z axis. All these movements were recorded during each stride, relative to vertical for the X and Z axes and horizontal for the Y axis. The range of motion in the sagittal plane was considered the full angular distance between the limb’s forward (protraction) and backward (retraction) movement. The abduction movement was measured while the limb was tilted outwards, and the adduction movement during inward tilting. Both these were measured during the strides’ swing phase, as defined by other authors [23,31]. The upward and downward movements (recorded on the Y axis) relative to the horizontal plane represented each stride’s hoof-off and hoof-on phases, respectively.

Although the sensor’s movements inside its case had been suppressed, other sources of incertitude could have been the inexact fitting of the devices, or their movement during the examination. To avoid these issues, the static acceleration was recorded in the standing horse for one leg at a time (while immobile and weight bearing), on each of the three axes of the accelerometer. When the device was fitted properly, the static acceleration (gravitational) was close to 0 on the X and Z axes, and 9.8 m/s^2^ on the Y axis. Based on these values, the positioning of the devices was readjusted as needed. Then the remaining errors were subtracted from the values recorded during the horse’s gait assessment. To limit this deviation, a vertical line was drawn both on the very middle of the devices’ case and the horses’ hooves (using a 0.1 mm tipped Statmark^®^™-pen (Weston, FL, USA), at half of the distance between the two margins of the devices, or the middle of the horses’ measured pastern circumference, respectively. The maximal possible uncertainty was estimated to be two degrees, caused by the inexact fitting of the devices’ angles relative to the anatomical ones. Three experienced veterinarians conducted a classical, subjective lameness assessment by observing the horses led by a helper at walk and trot, and in a straight line and circling, scoring each horse’s gait according to the lameness scale devised by the American Association of Equine Practitioners [32]. The final AAEP lameness score for each horse was decided upon by mutual agreement between the assessing clinicians and noted. At the beginning of this assessment, the horses were led along a straight line marked on the asphalt, at a walking pace, and at a constant pace. After a few steps, crossing a marking perpendicular to the straight line, five consecutive steps were recorded with the Lameness Detector 0.1, after which the devices were removed from the legs of the horses and the classical lameness exam continued under the observation of the three assessing veterinarians. The number of the steps to be recorded (five) was decided to be as small as possible to provide a rapid and easy possibility for the clinician for an initial lameness screening within the lameness assessment. The data provided by the Lameness Detector 0.1 were stored on the laptop for later evaluation and interpretation.

### 2.3. Data Processing and Statistical Analyses

The datasets provided by the Lameness Detector 0.1 device and processed by its desktop software were stored as motion graphs, also listing the acceleration values (m/s^2^) for each leg of the examined horses, on all three axes of the accelerometers, and for each of the five consecutive steps. The acceleration values (m/s^2^) were recorded in a separate Excel document, and, based on the motion graphs the steps, had were identified with their start and end values (Figure 4), allowing for the quantification of the data sent by the sensor from the start to the end of each step. As Figure 4 shows, the graphic representation of the sensor’s trajectory had similar shapes on the same axis at different steps.

The final values measured and analyzed by the Lameness Detector 0.1 were the impulses of the accelerometers, recorded by the Arduino Pro Mini microcontroller board, at every 10 milliseconds, which were then transformed and sent through the Bluetooth device to the Arduino software. The number of impulses was plotted on each axis during each step, and then the mean values of the impulse numbers for the five consecutive steps were calculated. These values were further analyzed using SPSS (version 17, 2010, www.spss.com (accessed on 20 March 2022) statistical software. For data comparison, the paired samples *t*-test was used, after testing the normality distribution by the Kolmogorov–Smirnov test. Spearman’s rank correlation coefficients (for non-parametric data) were calculated to verify the existence of correlations. The level of statistical significance was set at *p* < 0.05.

The present animal study protocol was approved by the Institutional Bio-Ethics Committee of the University of Agricultural Sciences and Veterinary Medicine of Cluj-Napoca (CBE decision no. 12/2014).

## 3. Results

The prevalence of each AAEP score decided by mutual agreement of the three assessing veterinarians is presented in Table 1.

For the statistical analysis, the mean values of the impulses were calculated for the five consecutive steps assessed in each lame leg and its contralateral sound leg.

### 3.1. Descriptive Statistics for the Mean Impulse Numbers Recorded in the Lame and Sound Contralateral Legs of the Assessed Horses

Table 2 shows an overview of the mean impulse values recorded in the assessed horses. The values have been arranged according to the accelerometers’ axes, and the lameness score of the horses

As Table 2 presents, the mean values of impulses on the X axis were higher in the lame legs, with maximal values in the lame hind-legs, even when the mean value was the same in the lame front- and hind-legs. The values increased with the lameness severity (higher lameness score on the AAEP scale). For all other axes, the impulses varied between 58.6 (minimum value recorded on the Z axis in sound forelegs) to 68 (maximum value recorded on the Y axis of sound forelegs). The minimum on the X axis in the sound legs fell within these limits but exceeded it considerably (86.4) in the lame forelegs.

### 3.2. Comparisons between the Mean Impulse Values Recorded in the Lame and the Sound Contralateral Legs of the Assessed Horses

The number of impulses recorded for each leg (both lame and sound) was calculated as a mean value for five consecutive steps and plotted separately for the fore and hind-legs, to be compared between the lame and the sound contralateral leg, for each lameness score, from 1/5 to 5/5 (Figure 5 and Figure 6). The mean value of impulses on the X axis was always higher in the lame leg than in the sound contralateral leg, both in the fore and hindlimbs. This difference increased with the lameness’ severity (Figure 5 and Figure 6) because of the gradual increase in the X axis-recorded impulses.

Without considering the lameness degree, the mean values of impulses for the five assessed steps were compared between the lame and sound contralateral forelegs (Table 3) and hind-legs (Table 4), on each of the accelerometers’ three axes.

The only statistically significant (*p* < 0.05) difference was found between the mean value of impulses recorded in the lame and sound forelegs on the X axis (Table 3).

Similarly, with the variation in the impulses’ mean value in the forelegs, the X axis showed a gradual increase in the lameness severity in the hind-leg (Figure 6). The difference in the mean impulse values between the lame and sound hind-leg was statistically significant (*p* < 0.05) not only on the X axis but also on the Z axis (Table 4).

As both Table 3 and Table 4 show, the power of the test was higher on the X axis (>0.89), regardless of if the front- or hind-legs were considered. For the horses with hind-leg lameness, the power of the test was low (0.5698) on the Z axis and very low (<0.15) for the rest of the instances. Thus, the null hypothesis (of the equality of the means) can be rejected for only the measurements on the X axis.

### 3.3. Correlations between the Mean Impulse Values Recorded in the Lame and the Sound Contralateral Legs of the Assessed Horses and Their Lameness Scores

To explore further the relationship between the mean value of impulses for the five steps recorded on the three axes of the accelerometers and the lameness severity in the studied horses, the Spearman’s rank correlation coefficients were calculated instead of linear correlations, because of the small sample size. Table 5 shows the results obtained for the horses with foreleg lameness and Table 6 shows those for the horses with hind-leg lameness.

The single statistically significant (*p* < 0.001) correlation coefficient found was between the impulses recorded on the X axis by the Lameness Detector 0.1 and lameness severity in the studied horses with foreleg lameness (Table 5). The correlation was positive, showing the increase in the impulses’ number with the more severe lameness diagnosed on the AAEP lameness scale.

In the same way, as for the forelegs, the hind-leg lameness showed a statistically significant (*p* < 0.001) positive correlation between the number of impulses recorded and lameness severity only on the X axis (Table 6). The correlation coefficients were negative on the Y and Z axes, and the number of impulses decreased as the lameness scores increased.

## 4. Discussion

The walk is a four-beat symmetrical gait in horses, displaying the alternate movement of the contralateral limb pairs and having the four footfalls symmetrically spaced in time [6], provided that the horse’s locomotion is sound. As the AAEP [32] describes, lameness is a change in the horse’s gait, usually caused by pain, or as a result of a mechanical restriction on movement. Thus, the ‘lameness’ term is not limited to describing obvious limping, but it includes subtle gait changes too, or even the decreased ability or willingness to perform. As van Weeren et al. [33] state, the lame–sound dichotomy is traditionally rooted in the equestrian language. It bears a strong underlying general perception that lame horses are unfit to perform, and their continued use impairs their welfare. Thus, lameness is a pathological condition, a clinical problem that needs timely diagnosis, from its earliest and slightest signs, and proper treatment to avoid further worsening and permanent impairment of the animal’s locomotion and, possibly, overall life quality. To describe the gait of horses with conformational defects, training or shoeing errors, and unequal muscle development, sometimes different terms are used (‘uneven/rough/irregular/abnormal gait’, ‘gait asymmetry’, or ‘out-of-balance locomotion’), and these animals have been tacitly considered as being rather lame too, waiting for a more definitive or thorough diagnostic to prove it [33]. As van Weeren et al. state [33], lameness is similar to most non-emergency and not immediately life-threatening clinical conditions: the more thorough an examination is performed, the more chances are for positive diagnoses. Thus, equine medicine has continuously improved, for centuries, to find the best possible lameness examination procedures. The paradox of this situation happens currently, with the emergence of very sensitive objective gait analysis possibilities in horses when veterinary medicine faces an unexpected concern and a possible need to redefine equine lameness [33,34] to avoid its over-diagnosing. Thus, the integration of gait analysis technology within equine clinical practice is an ongoing process that warrants careful monitoring [35]. Ironically, the increased sensitivity in detecting slight gait asymmetries which could be difficult to observe in a classical lameness examination to the human eye is considered both the strength and weakness of modern devices used for objective gait analysis. Although this sensitivity, exactness, and lack of bias are specifically pursued, to allow for a precise diagnosis at certain ‘threshold values’, the opposing concern is that if an asymmetry is so mild that it is difficult to detect, its clinical relevance cannot be safely certified [34]. Considering all these aspects, our Lameness Detector 0.1 device was intended to be a screening tool to improve the subjective lameness assessment. Used at the beginning of the classical observation, it is meant to increase the practitioner’s attention to perform a more thorough and detailed examination when the number of the recorded impulses is higher than normal.

In our study, the number of impulses recorded by the Lameness Detector 0.1 increased gradually with both fore- and hindlimb lameness severity. The most relevant increase was noted in the impulses provided by the accelerometer’s X axis. The explanation of this finding lies in the mechanical dynamics of the horse’s gait when walking. Considering the three phases of the step (the hoof-off moment, the swing phase, and the hoof-on moment) most of the movement detected by the sensor on the horizontal plane took place during the step’s middle phase (swing). The vertical axis of the sensor, the Y axis, recorded the dynamic acceleration during the other two phases (the hoof-off and hoof-on moments) the upward and downward movements made by the horses’ legs, and on the Z axis, the lateral (and medial) movements were registered, alongside any deviation from the median axis of the horses’ limbs. The number of impulses captured at the hoof-off and hoof-on moments of the step (Y axis) differed between the lame limb and the control healthy limb because, due to pain, the amplitude of the step decreased in the lame limb. This result might have been caused by the hesitancy of the horse to place the lame limb on the ground, generating a slight swinging recorded on the Z axis.

The analysis of the mean impulse values indicated identical dynamics for the sound legs, but significant differences between the sound and lame legs on the X axis. For both fore- and hindlimbs, acceleration values increased for the healthy limbs and decreased for the lame limbs. Previous studies have also indicated that vertical force peak and impulse have high relevance in assessing lameness severity [36,37].

A sensor-based device currently commercialized under the Lameness Locator name has been studied and validated to detect lameness [38]. This device analyzes the fore- and hind-leg movements by measuring the pelvic movement asymmetry and the differences between the maximum and minimum positions of the pelvis between the right and the left portion of the step. The differences between our device and the Lameness Locator are major. The latter is equipped with two accelerometers and a gyroscope. The accelerometers have to be fixed on the head and pelvis of the horse, and the gyroscope on its right fore- or hindlimb. In this set-up, the accelerometers measure the acceleration of the trunk in a vertical plane, and the gyroscope is responsible for detecting the asymmetry between the lame and the contralateral sound limb. The acceleration of the trunk is then converted into the position, and the signal is decomposed into harmonic and random components of motion. Our Lameness Detector 0.1 had four gravitational sensors, accelerometers that recorded limb acceleration at each step, and emitted pulses during the entire step duration on all three axes. Both acceleration values and pulse frequency were used to quantify the time spent at each step, and to interpret the movements performed by the lame leg, with different lameness severity, compared to the healthy limb. As opposed to the Lameness Locator, the Lameness Detector 0.1 was able to assess lameness on all four limbs simultaneously.

Based on the number of transmitted impulses, lameness scores can also be assigned. Another wireless inertial measurement system (EquiMoves) has recently been described and validated compared to an optical motion capture system [23]. The results of the EquiMoves testing indicate a good agreement and a low level of bias between the two systems in terms of locomotor parameters of both sagittal (protraction, retraction, sagittal range of motion) and coronal (adduction, abduction, coronal range of motion) planes [23].

The Inertial sensor-based examination cannot completely replace the subjective clinical assessment, but it can support it by enhancing and validating the accuracy and reliability of subjective results. Considerable literature data indicate the versatility of inertial sensor-based wireless devices in equine gait assessment [38,39,40,41] and their utility in removing bias from clinical decision-making, aiding the less-experienced clinician, and providing evidence for the lameness diagnostic. However, as the AAEP [32] highlights, for equine veterinarians, lameness diagnosis and treatment are both a science and an art, requiring a solid understanding of all the participating structures in the horses’ locomotion on the one hand, and needing adaptation in response to changing conditions, horse types, uses, personalities, and owner needs, on the other hand. This way, all devices aiding the veterinary activity are valuable in their own regard, without replacing the knowledge, experience, and even talent needed to interpret the quantitative results properly.

## 5. Conclusions

The number of impulses recorded on the Lameness Detector 0.1’s incorporated accelerometer’s X axis increased with the severity of the studied horses’ lameness in both their fore- and hind-legs, and was significantly higher than those recorded on the Y and Z axes. Thus, the values of the X axis proved to be the most relevant for the intended use of the Lameness Detector 0.1. In the light of the obtained results, further validation and standardization of the Lameness Detector 0.1 is considered warranted by our team to develop an affordable and easy-to-use, readily available device that can support, as a screening tool in the first instance, the equine veterinarians in their routine activity of assessing gait performance and lameness in horses.

## Figures and Tables

**Figure 1 sensors-22-07082-f001:**
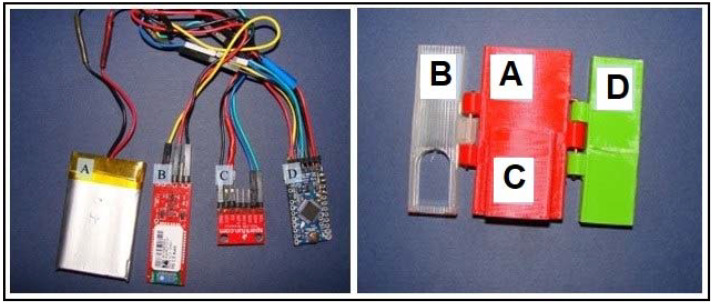
The components of a Lameness Detector 0.1 device (**left panel**): accumulator (**A**), Bluetooth (**B**), accelerometer (**C**), Arduino board (**D**), and their case (**right panel**) with the appropriate boxes for each component, labeled accordingly ((**A**) for the accumulator, (**B**) for the Bluetooth, (**C**) for the accelerometer, and (**D**) for the Arduino board, respectively).

**Figure 2 sensors-22-07082-f002:**
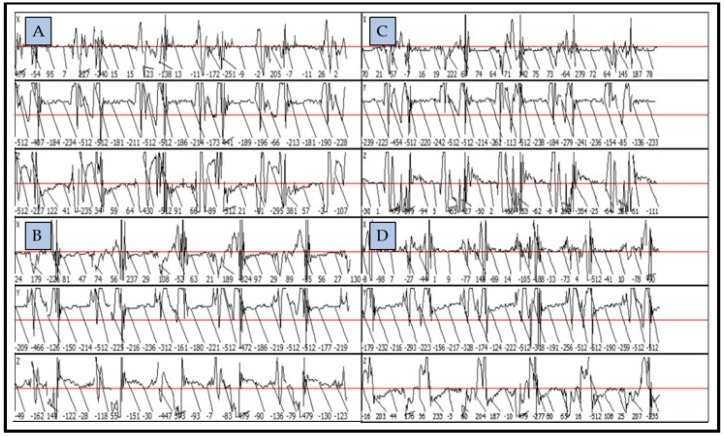
Graphic representation of the gait analysis as recorded and processed by the Lameness Detector 0.1 for the four legs of a horse: right fore (**A**), right hind (**B**), left fore (**C**), and left hind (**D**), showing both the values of acceleration (m/s^2^) and trajectory of the accelerometer on its three axes.

**Figure 3 sensors-22-07082-f003:**
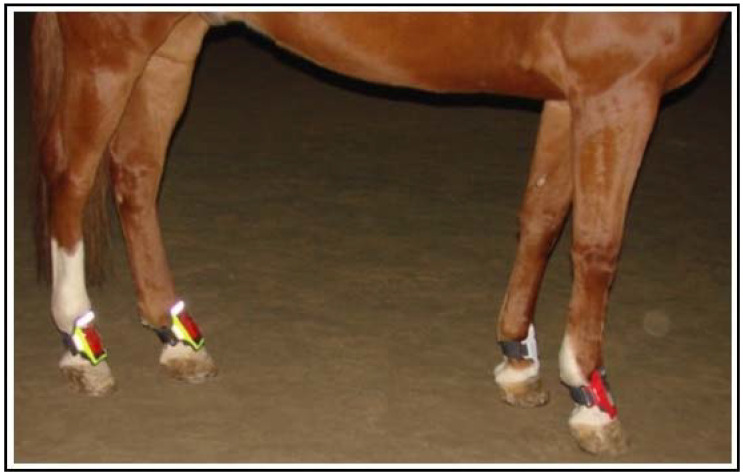
Illustration of correct fitting of the Lameness Detector 0.1 system with adjustable straps to the dorsal aspect of a horse’s pasterns.

**Figure 4 sensors-22-07082-f004:**
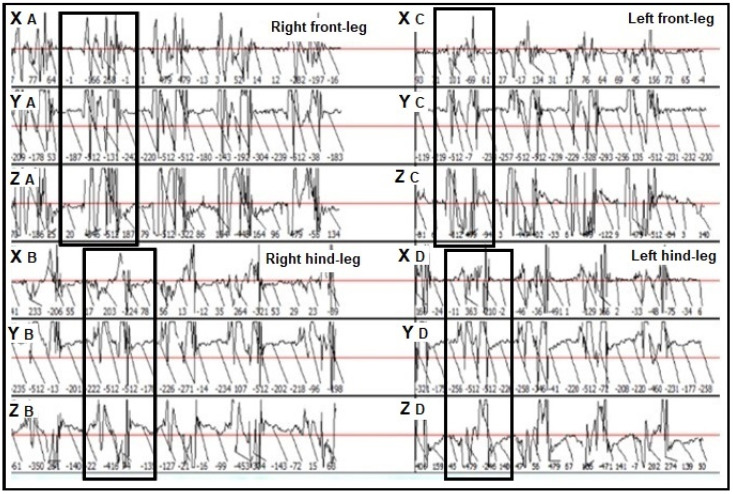
Step identification (rectangles) on the graphs constructed by the Lameness Detector 0.1 software. XA: protraction–retraction movement of the right front-leg recorded on the X axis; YA: upward–downward movement of the right front-leg recorded on the Y axis; ZA: abduction–adduction movement of the right front-leg recorded on the Z axis; XB: protraction–retraction movement of the right hind-leg recorded on the X axis; YB: upward–downward movement of the right hind-leg recorded on the Y axis; ZB: abduction–adduction movement of the right hind-leg recorded on the Z axis; XC: protraction–retraction movement of the left front-leg recorded on the X axis; YC: upward–downward movement of the left front-leg recorded on the Y axis; ZC: abduction–adduction movement of the left front-leg recorded on the Z axis; XD: protraction–retraction movement of the left hind-leg recorded on the X axis; YD: upward–downward movement of the left hind-leg recorded on the Y axis; ZD: abduction–adduction movement of the left hind-leg recorded on the Z axis.

**Figure 5 sensors-22-07082-f005:**
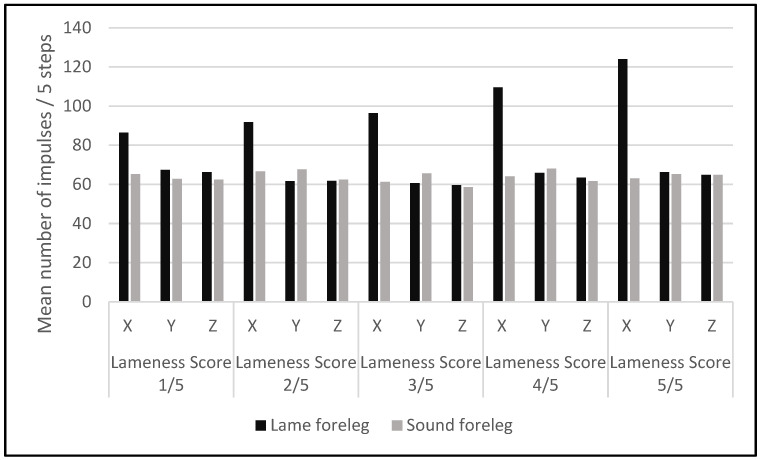
Mean numbers of impulses for five steps, recorded by the Lameness Detector 0.1 in the sound and lame forelegs with different lameness scores.

**Figure 6 sensors-22-07082-f006:**
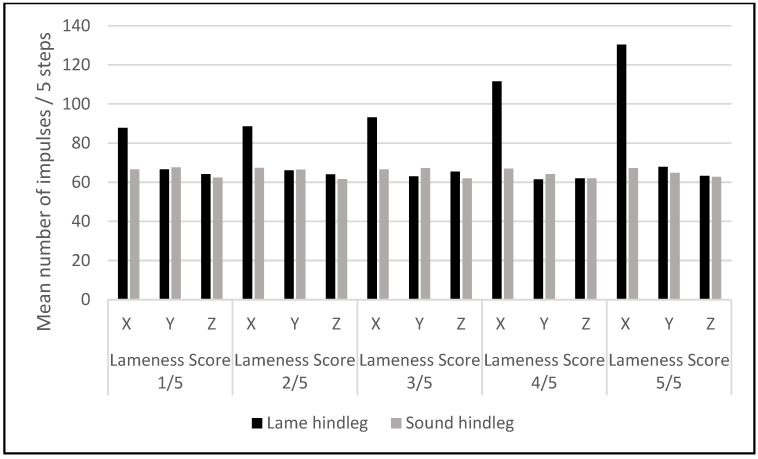
Mean numbers of impulses for five consecutive steps recorded by the Lameness Detector 0.1 in the sound and lame hind-legs with different lameness scores.

**Table 1 sensors-22-07082-t001:** The diagnosed AAEP scores and their prevalence in the studied horses (*n* = 10), according to their lame leg.

AAEP Score	Right Fore	Left Fore	Right Hind	Left Hind	AAEP [28] Guidelines for Lameness Grading System
1/5	1	0	0	1	Lameness is difficult to observe, and it is not consistently apparent, regardless of the circumstances
2/5	0	1	0	1	Lameness is difficult to observe at a walk or when trotting in a straight line but is consistently apparent under certain circumstances
3/5	1	0	1	0	Lameness is consistently observable at a trot under all circumstances
4/5	1	0	0	1	Lameness is obvious at a walk
5/5	0	1	1	0	Lameness produces minimal weight bearing in motion and/or at rest or a complete inability to move

**Table 2 sensors-22-07082-t002:** Descriptive statistics of the mean impulse values recorded by the Lameness Detector 0.1 in the studied horses (*n* = 10).

Lameness Scores	Sound Foreleg			Sound Hind-Leg			Lame Foreleg			Lame Hind-Leg		
	X Axis	Y Axis	Z Axis	X Axis	Y Axis	Z Axis	X Axis	Y Axis	Z Axis	X Axis	Y Axis	Z Axis
1/5	65.2	62.8	62.4	66.6	67.6	62.4	86.4	67.4	66.2	87.8	66.6	64.2
2/5	64	63.6	63.4	67.4	66.4	61.6	91.8	61.6	61.8	88.6	66	64
3/5	61.2	65.6	58.6	66.6	67.2	62	96.4	60.6	59.6	93.2	63	65.4
4/5	64	68	61.6	67	64.2	62	109.6	65.8	63.4	111.6	61.4	62
5/5	63	65.2	64.8	67.2	64.8	62.8	124	66.2	64.8	130.4	67.8	63.3
MIN	61.2	62.8	58.6	66.6	64.2	61.6	86.4	60.6	59.6	87.8	61.4	62
MAX	65.2	68	64.8	67.4	67.6	62.8	124	67.4	66.2	130	67.8	65.4
Mean	63.5	65	62.2	67	66	62.2	102	64.3	63.2	102	65	63.8
Median	64	65.2	62.4	67	66.4	62	96.4	65.8	63.4	93.2	66	64
STDEV.S	1.49	2.01	2.32	0.36	1.49	0.46	15.2	3.02	2.57	18.4	2.66	1.25
SEM	0.67	0.9	1.04	0.16	0.66	0.2	6.78	1.35	1.15	8.24	1.19	0.56

MIN: minimum; MAX: maximum; STDEV.S: standard deviation of the sample; SEM: standard error of the mean.

**Table 3 sensors-22-07082-t003:** Comparison between the mean number of impulses recorded on the accelerometers’ axes for the five assessed steps in the lame and sound contralateral foreleg of the studied horses (*n* = 5).

Axis	In the Lame Foreleg		In the Sound Foreleg		Comparison of the Means		Effect	The Power of the Test
	m	σ	m	σ	t(4)	*p*	d (Cohen)	
X	101.64	15.16	63.48	1.49	5.449	0.006 *	2.437	0.9771
Y	64.32	3.02	65.04	2.01	0.441	0.682	0.197	0.0638
Z	63.16	2.57	62.16	2.32	1.110	0.329	0.496	0.1391

m: mean number of impulses; σ: standard deviation. * For *p*-values less than 0.05 the difference between the means was considered statistically significant.

**Table 4 sensors-22-07082-t004:** Comparison between the mean number of impulses recorded on the accelerometers’ axes for the five assessed steps in the lame and sound contralateral hindleg of the studied horses (*n* = 5).

Axis	In the Lame Hind-Leg		In the Sound Hind-Leg		Comparison of the Means		Effect	The Power of the Test
	m	σ	m	σ	t(4)	*p*	d (Cohen)	
X	102.32	18.42	66.96	0.36	4.323	0.012 *	1.933	0.8908
Y	64.96	2.66	66.04	1.49	0.885	0.426	0.396	0.1065
Z	63.84	1.23	62.16	0.46	2.822	0.048 *	1.262	0.5698

m: mean number of impulses; σ: standard deviation. * For *p*-values less than 0.05 the difference between the means was considered statistically significant.

**Table 5 sensors-22-07082-t005:** Spearman’s rank correlation coefficients and their significance showing the relationship between the main value of the impulses and lameness scores in the horses with a lame foreleg (*n* = 5).

Axis	In the Lame Foreleg		In the Sound Foreleg	
	*ρ*	*p*	*ρ*	*p*
X	1.000 *	<0.001	−0.616	0.269
Y	−0.100	0.873	0.700	0.188
Z	−0.100	0.873	0.200	0.747

* Correlation coefficient significant at *p* < 0.05.

**Table 6 sensors-22-07082-t006:** Spearman’s rank correlation coefficients and their significance showing the relationship between the main value of impulses and lameness scores in the horses with a lame hind-leg (*n* = 5).

Axis	In the Lame Hind-Leg		In the Sound Hind-Leg	
	*ρ*	*p*	*ρ*	*p*
X	1.000 *	<0.001	0.308	0.614
Y	0.000	1.000	−0.800	0.104
Z	−0.600	0.285	0.359	0.553

* Correlation coefficient significant at *p* < 0.05.

## Data Availability

The data supporting the reported results can be found through C.M.C. and M.A.R., available at request.

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
