# Peer review of "Development of a Novel Approach for Detection of Equine Lameness Based on Inertial Sensors: A Preliminary Study"

_sensors, 2022, doi:10.3390/s22187082_

Round 1

Reviewer 1 Report (New Reviewer)

Authors describe a inertial-based sensor system used for the analysis of horses with different lameness degree. The authors also designed a software for the data processing. Paper is well structured and the aims are in line with the journal topic. However, some improvements should be considered before the publication.

1. The main issue is the almost total absence of previous studies on the same topic. This aspect is essential to understand the novelty of the paper and to compare the obtained results with respect other methodologies. The introduction should be focused on the lameness and not in general on the instruments used for movement analysis. 

2. Due to small sample size, authors should provide power analysis of the applied statistics. 

3. Authors should indicate possible uncertainty source due to the sensor placement.

4. Since authors talk about angle in specific plance, they should report if a functional calibration has been performed to align sensor axes with the anatomical one. In addition, more details on angle extraction should be added.

5. Statics should be better discussed, highligthing the main aims. 

6. Authors in the abstract used the term "reliable", no reliability analysis has been conducted in the study. 

Author Response

Review 1

Thank you for your review and comments, and for your time and involvement in improving our paper. Please find below our answers addressing the problems highlighted during the review process.

  1. The main issue is the almost total absence of previous studies on the same topic. This aspect is essential to understand the novelty of the paper and to compare the obtained results with respect to other methodologies. The introduction should be focused on the lameness and not in general on the instruments used for movement analysis.

Thank you for your comment. We inserted a brief introduction about equine lameness but keeping in mind the scope of the Journal and the recommendations of our other reviewer, we kept this aspect short. Also, we tried to reference those movement analysis methodologies and instruments that are used in equine gait analysis (see references 6, 7, 8, 9, 10, 11, 22-24, 25-28, 30, 35-38). 

  1. Due to the small sample size, authors should provide power analysis of the applied statistics.

Thank you for your comment. The results of the power analysis have been added to Table 3 and Table 4 (formerly Table 2 and Table 3), with a brief explanation (below Table 4), as follows:

“As both Tables 3 and 4 show, the power of the test is higher on the X axis (>0.89), regardless if the front- or hind-legs have been considered. For the horses with hind-leg lameness the power of the test is low (0.5698) on the Z axis and very low (<0.15) for the rest of the instances. Thus, the null hypothesis (of the equality of the means) can be rejected for only the measurements on the X axis.”

  1. Authors should indicate possible uncertainty source due to the sensor placement.

Thank you for your valuable insight and comment. We hope we succeeded to indicate properly the uncertainty sources we have thought of while performing the study. We added our explanations in this regard to the Materials and Methods section, as follows:

“In order to prevent the movements of the accelerometers in their cases, which could have introduced errors in the measurements, these had been glued inside the custom-tight made cases (using Loctite® Super Glue Ultra Gel Control™).”

“Although the sensor’s movements inside its case had been suppressed, other sources of incertitude could have been the inexact fitting of the devices or their movement during the examination. In order to avoid these issues, the static acceleration was recorded in the standing horse for one leg at a time (while immobile and weight-bearing), on each of the three axes of the accelerometer. When the device was fitted properly, the static acceleration (gravitational) was close to 0 on the X and Z axes, and 9.8m/s2 on the Y axis. Based on these values the positioning of the devices was readjusted as needed, then the remaining errors had been subtracted from the values recorded during the horse’s gait assessment. To limit this deviation a vertical line was drawn both on the very middle of the devices’ case and the horses’ hooves (using a 0.1 mm tipped Statmark®™-pen, at half of the distance between the two margins of the devices, or the middle of the horses’ measured pastern circumference, respectively). The maximal possible uncertainty was estimated to be less than two degrees, caused by the inexact fitting of the devices’ angles relative to the anatomical ones.”

  1. Since authors talk about angle in specific plane, they should report if a functional calibration has been performed to align sensor axes with the anatomical one. In addition, more details on angle extraction should be added.

Thank you for your very valuable comment. The mentioned issue has been addressed in the manuscript’s Materials and Methods section (see also the above discussion).

  1. Statics should be better discussed, highlighting the main aims.

Please explain to us better what you meant by ‘statics’.

  1. Authors in the abstract used the term "reliable", no reliability analysis has been conducted in the study. Thank you for your comment. No reliability analysis has been made; indeed, thus we changed the wording of the abstract, as follows:

“By recording the impulses on three axes of the incorporated accelerometer in each leg of the assessed horse, then processing the data by a custom-designed software, the device proved its usefulness in lameness identification and severity scoring.”

Reviewer 2 Report (New Reviewer)

The authors of this paper aim to propose a new inertial sensor and an approach enabling to detect lameness in horses through respective gain analysis.

The idea is interesting and possibly of some impact (the reviewer is no expert in the field of lamness of horses). On the strong points authors do develop a real-life system which is used in the real filed (i.e. real horses) while they are using real experts considered the opinion of which is used as the ground through and they are following a standardized approach regarding respective lameness scale introduced by the AAEP. These are aspects increasing objectivity and validity of the methodology presented and observations.

The use of English is mediocre and carefully review is certainly required by native English speaking and writing person. E.g use battery instead of accumulator os an example.

But in the reviewer’s opinion there are several structural and conceptual as well as technical issues and aspects that rains significant concerns.

In section 2.1 the developed system is presented. The idea is clear but hardly innovative while the reason of creating a new device is not clear. There are plenty low cost, open platforms of very small form factor that can do the work quite reliably and they are actually used in such scenarios like gait analysis or movement pattern monitoring etc. while they are also 9DoF such devices. So the contribution of this device is quite questionable. No discussion is made on what are the advantages. Considering that Journal is entitled Sensors this should be the focus of the paper actually. On the contrary this section is the only section clearly related to sensors. So this is a negative aspect considering the journal at which it is submitted.

On the other hand the sw architecture/application could be something interesting but should be also compared to the competition (many relative platforms come with very flexible and relatively open SWs) and could more well fitted to a computer science journal.

A technical question concerns the selection Bluetooth as the main communication technology. Considering the limited range of BT interfaces, considering open fields or relative areas probably WiFi would be a better choice.

Indicative all in one solution to be checked included but are not limited to https://store-usa.arduino.cc/products/arduino-nano-33-ble  https://www.ti.com/tool/TIDC-CC2650STK-SENSORTAG https://shimmersensing.com/product/shimmer3-imu-unit/ and many more.

With respect to lines 195-200. It would be very helpful if authors provide images indicating clearly the gait analysis of horses of both front and back legs. Elaborate on what is normal and what is out of range regarding all movements and considering ranges.

With respect to 202-204. Nice indication. A table or matrix is needed indicating what the experts reported as the ground truth compared to which the proposed solution is evaluated.

As it is Figure 4 is not very useful although important.  Authors need to be much more detailed on what is shown and where it is shown in the graphs. What is the right hand side. The analysis is based on what is "shown". This is not adequate for a scientific paper. Quantification measurements of differences which need to be 0 etc. are required. Overall the Figure 4 is very important but at current form abstract and sketchy.

Also elaborate on how many steps did they make.

Concerning table 1. I would suggest to use graphs rather than absolute numbers to compare results and show important outcomes. Also please elaborate referring to specific number when you try to deduce an observation. As it is not it very difficult to get something out of this comparison. Also deviations together with mean values are critical to be depicted graphically and analyzed as well.

Concerning Table 2. what is ρ? How are t and p calculated and what are their qualitative significance?

With respect to lines 318-321 not clear where this is shown. what is the meaning of positive and negative results ?

With respect to table 5 analysis. Elaboration and in-depth analysis is needed as to the explanation of the measurements. Elaborate on all cases.

With respect to line 389. This basically the only reference to the sensor itself at the discussion section ? Considering that the paper proposes a new sensor and is submitted at a Sensor Journal this is very limited.

Apart from the rest of the observation it is the reviewer's opinion that there is a significant issue regarding how the measurements are taken into consideration to decide if there is lameness and to what degree. The statistics are based on absolute numbers considering a very small sample. And the comparison is done after the experts have decided where the lameness resides and to what degree for each sample. However, since these results are based on specific absolute number their can be somehow considered objective for these samples but to for all horses that could/can be tested.

In other words, if a new horse comes for examination, how would someone draw conclusion based on the measurements already done. Without taking into consideration the height, weight, age and other parameter (e.g. lameness could exist in two legs) etc. It can not be objectively done solely by these numbers.

That is why in such cases e.g. human gait analysis, ML/AI models are trained and then tested in a high number of users to objectively and accurately detect and categorize and observation. It is the reviewer opinion that this approach should be followed here as well. And then the work is basically computer data science.

The percentage of the paper dedicated to the analysis of the horse lameness with respect to the sensors and statistical analysis is quite uneven, making the paper somehow borderline as it regards being out of scope, especially if we consider the journal to which it is submitted.

Based on the previous comments the paper offers limited contribution regarding the sensors and data analysis and there are several issues clearly presented raising concerns. Therefore even through the application domain is interesting it is the reviewer opinion that it should not be accepted for publication

Author Response

Review 2

Thank you for your review and comments, and for your time and involvement in improving our paper. Please find below our answers addressing the problems highlighted during the review process.

The use of English is mediocre and carefully review is certainly required by native English speaking and writing person. E.g use battery instead of accumulator as an example.

Thank you for your comment. The English language of the manuscript has been reviewed. Regarding the ‘accumulator’ term, it was intentionally used to respect the manufacturer’s product description, but we have changed it to ‘rechargeable battery’.

In section 2.1 the developed system is presented. The idea is clear but hardly innovative while the reason of creating a new device is not clear. There are plenty low cost, open platforms of very small form factor that can do the work quite reliably and they are actually used in such scenarios like gait analysis or movement pattern monitoring etc. while they are also 9DoF such devices. So the contribution of this device is quite questionable. No discussion is made on what are the advantages. Considering that Journal is entitled Sensors this should be the focus of the paper actually. On the contrary this section is the only section clearly related to sensors. So this is a negative aspect considering the journal at which it is submitted.

Thank you for your comment. We added our explanations to the Discussions section as follows:

“Taking into account all these aspects, our Lameness detector 0.1 device was intended to be a screening tool, to improve the subjective lameness assessment. Used at the beginning of the classical observation, it is meant to increase the practitioner's attention to perform a more thorough and detailed examination when the number of the recorded impulses would be higher than normal.”

Indeed, there are several other similar devices on the market already, but each of them has some weak points compared to ours. For instance:

  • Only a lameness diagnostic using the EquiMotion costs 220 USD (https://wallabyhill.com.au/product/equimotion-clinic/);
  • According to Keegan, the current price for a Lameness Locator

and the online training is 15,000 USD (https://www.americanfarriers.com/articles/682-product-innovation-lameness-locator-moves-equine-veterinary-technology-a-step-ahead?v=preview);

By contrast, our device can be built for less than 200 USD per foot.

  • The powerful Bluetooth component built in our device makes the data recording possible even in field conditions in remote areas, outside the equine-clinic settings;
  • Our device system is easier to attach to the equine body, without the need of placing components on the horse’s head and/or rump.
  • Our software is open-source, downloadable (from here: https://github.com/mihaioltean/lameness-detector), and thus adaptable to the user’s needs.

On the other hand, the sw architecture/application could be something interesting but should be also compared to the competition (many relative platforms come with very flexible and relatively open SWs) and could more well fitted to a computer science journal.

Thank you for your comment. We did not include the information in the manuscript, but our software is also open-source, here: https://github.com/mihaioltean/lameness-detector.

A technical question concerns the selection Bluetooth as the main communication technology. Considering the limited range of BT interfaces, considering open fields or relative areas probably WiFi would be a better choice. Indicative all in one solution to be checked included but are not limited to https://store-usa.arduino.cc/products/arduino-nano-33-ble https://www.ti.com/tool/TIDC-CC2650STK-SENSORTAG https://shimmersensing.com/product/shimmer3-imu-unit/ and many more.

Thank you for your valuable comment. We decided on Bluetooth data transmission for the same main reason for which the all-in-one devices use the same interface: the considerably lower power consumption as compared to the ‘true’ wireless connections. As per our knowledge, Arduino nano 33 BLE’s Bluetooth transmission range is only a few meters (see: https://forum.arduino.cc/t/ble-very-weak-signal/631751/9 ). Our device has a range of 100m.

With respect to lines 195-200. It would be very helpful if authors provide images indicating clearly the gait analysis of horses of both front and back legs. Elaborate on what is normal and what is out of range regarding all movements and considering ranges.

Lines 195-200 describe the range of motions of the equine gait to explain further the previous two sentences describing the accelerometers’ positioning on the horses’ feet, and what was recorded on each axis of the accelerometers. Our devices do not measure the spatial range of these movements (the length of protraction, retraction, abduction, adduction, or the height of the upward feet-movement) to try to establish what is out of range in those movements (as being out of spatial range does not necessarily indicate lameness), but the acceleration during each phase of each step, and also the duration of each phase (possible to be calculated as the sensor emits one impulse per ten milliseconds). However, the differences in the recorded values when the movements of different legs are compared indicate possible problems, and this is the functional principle of our screening device.

With respect to 202-204. Nice indication. A table or matrix is needed to indicate what the experts reported as the ground truth compared to which the proposed solution is evaluated.

Thank you for your comment. The requested table has been introduced in the Results section (Table 1), to replace the textual description of the AAEP scores’ prevalence in the assessed horses, and we added the AAEP protocol too.

As it is Figure 4 is not very useful although important. Authors need to be much more detailed on what is shown and where it is shown in the graphs. What is the right hand side. The analysis is based on what is "shown". This is not adequate for a scientific paper. Quantification measurements of differences which need to be 0 etc. are required. Overall the Figure 4 is very important but at current form abstract and sketchy. 

Thank you for your comment and suggestions. Figure 4 and its caption have been modified accordingly.

Also elaborate on how many steps did they make.

Thank you for your comment. This information has been included in the Abstract, then in the manuscript too (five steps), mentioning our motivation for choosing this low number of steps, in the Materials and Methods section, as follows:

“After a few steps, at crossing a marking perpendicular on the straight line, five consecutive steps had been recorded with the Lameness detector 0.1, after which the devices had been removed from the legs of the horses and the classical lameness exam continued under the observation of the three assessing veterinarians. The number of the steps to be recorded (five) was decided to be as small as possible, in order to provide a very quick and easy possibility for the clinician for an initial lameness screening within the lameness assessment.”

Concerning table 1. I would suggest to use graphs rather than absolute numbers to compare results and show important outcomes. Also please elaborate referring to specific number when you try to deduce an observation. As it is not it very difficult to get something out of this comparison. Also deviations together with mean values are critical to be depicted graphically and analyzed as well.

Thank you for your comment. Table 2 (formerly Table 1) presents the descriptive statistical parameters of the processed data. Although we mentioned that the mean values of impulses on the X axis were higher in the lame legs than the sound contralateral legs, that was not referring to the statistical analysis of comparison (no statistical significance was involved or mentioned). representing in graphs all the data presented in the table would take too much space and would not be justified in our opinion.

Concerning Table 2. what is ρ? How are t and p calculated and what are their qualitative significance?

Thank you for your comment and observation. Table 3 (formerly Table 2) and Table 4 (formerly Table 3) have been corrected and completed.

With respect to lines 318-321 not clear where this is shown. what is the meaning of positive and negative results?

Thank you for your comment. This information has been added.

With respect to table 5 analysis. Elaboration and in-depth analysis is needed as to the explanation of the measurements. Elaborate on all cases.

Thank you for your comment. The tables 5 and 6 (formerly 4 and 5) present the correlation calculation’s results, showing the relationship between the mean value of impulses recorded during five steps, on each of the three axes of the accelerometers, and the lameness severity in the studied horses.

With respect to line 389. This basically the only reference to the sensor itself at the discussion section?

Considering that the paper proposes a new sensor and is submitted at a Sensor Journal this is very limited.

Thank you for your comment. Although not strictly about the characteristics of one single sensor (but of sensor systems) the comparison of our device to the commercially available sensor systems used to assess equine gait may fit the Journal’s scope too.

Apart from the rest of the observation it is the reviewer's opinion that there is a significant issue regarding how the measurements are taken into consideration to decide if there is lameness and to what degree. The statistics are based on absolute numbers considering a very small sample. And the comparison is done after the experts have decided where the lameness resides and to what degree for each sample. However, since these results are based on specific absolute number their can be somehow considered objective for these samples but to for all horses that could/can be tested.

In other words, if a new horse comes for examination, how would someone draw conclusion based on the measurements already done. Without taking into consideration the height, weight, age and other parameter (e.g. lameness could exist in two legs) etc. It can not be objectively done solely by these numbers.

That is why in such cases e.g. human gait analysis, ML/AI models are trained and then tested in a high number of users to objectively and accurately detect and categorize and observation. It is the reviewer opinion that this approach should be followed here as well. And then the work is basically computer data science.

Thank you for your comments. Supplementary explanations have been added in several places of the manuscript, emphasizing that our device is not meant to replace the classical lameness examination, but to complete it, and that we propose it as a screening tool meant to attract the clinician’s attention to perform a more thorough examination of the animals found with gait-problems indicated by the device (impulse numbers on the accelerometer’s X axis). As in all computer- and artificial intelligence-assisted domains, these solutions are not meant to replace the experience and knowledge of the specialist, but to enhance their professional performance.

Round 2

Reviewer 1 Report (New Reviewer)

Authors properly answered to my previous  comments 

Author Response

Thank you for your time and for your valuable comments and corrections to our manuscript.

Reviewer 2 Report (New Reviewer)

I would like to thank the authors for the taking the time and putting the effort to elaborate on several aspects of the paper indicated in the initial review.

However, at the end of the day none of the weak technical points are actually addressed and the additional text, although helpful, it is not enough to actually correct them.

There I regret to inform that no significant enhancement or improvement has been made from the previous review.

Author Response

Thank you for your time and for your valuable comments and corrections to our manuscript.

This manuscript is a resubmission of an earlier submission. The following is a list of the peer review reports and author responses from that submission.

Round 1

Reviewer 1 Report

Review animals-1685757

The paper is possibly potential but a lot of the text should be re-written and clarified. It should be submitted better to Sensors, because of its technical type. 

However, I attached my comments:

1. introduction is based on old literature, concentrated on human science. The paper should be based on such papers as 32-34 and more. More papers can be cited here (see Sensors). All the info given in the introduction is true. However, it is not up-to-date info. The introduction should present the state of the art and the problems that the authors would like to solve. Why do they start to produce new equipment if they are others? (cited in discussion for example). The accelerometers are used for many years in horses also. Please provide the information on what will be new and better in your system. New knowledge starts in the introduction from line 97 (so too late). The introduction can start from the L 78 without losing too much info. The knowledge above (up to 97 can be mentioned but in a few sentences).

2. material and methods - the technical device should be better reviewed in more technical Sensors. The study is based on a hard surface. What about a soft one? Horses are mixed, which may influence the results. As it looks like from figure 3 the devices influence the horse's range of motion. Their positioning is not described in detail in the text. Did you check it anyway?   

L 267-279 – it is not clear what value is measured and analyzed. Please be as specific as possible. 

L 278 – does it mean that they agree? Not clear. Ethical statement not mentioned here in the text or the last part of the paper.

3. results – table 1 is described, and tables 2-5 are not. What is it m? and others? It should be clear from the tables themselves.

4. discussion – there are many interesting parts, however, the language should be more formal and scientific. Needs revision.

L 380-382 – interesting part, citations needed.

L 409 – please use proper wording (Back and Clayton 2013) for scientific comparison

 L 447 It would be interesting if your data will be also validated by another objective system, not only the subjective judgment of veterinarians.

5. conclusions – your preliminary results should be concluded. The information that the system will be validated is not the result.  

Author Response

Introduction – old literature, keep 32-34 and cite more form Sensors – Thank you for your comment. This aspect had been addressed.

Why do they start to produce new equipment if there are others? (Cited in discussion for example). What will be new and better in your system? – Thank you for your questions. Our device is meant as a very affordable alternative to the already existing ones, efficient enough to be used as a screening dispositive. The costs of the device are not at all limitative for even veterinarians at the beginning of their career (who we consider would benefit by using it), and the simplicity of both collecting and interpreting the results allows a good time-management of lameness assessment at the veterinary practice or even in field conditions. Thus, our device is meant to ensure rapid detection of gait abnormalities, to attract the attention of the evaluating practician to prompt her or him to further broaden and detail the lameness diagnosis procedure, when the device indicates a certain degree of lameness in the horse.

Start from L97 and mention the rest before in a few sentences - Thank you for your advice, this issue had been addressed.

Material and methods
The technical device should be better reviewed in more technical Sensors. – Thank you for your observation. Although technical, we consider our paper’s intended audience the veterinarians at the beginning of their career and who are interested in studying more about equine lameness. Considering the theme of the present special issue (Lameness assessment), our intended audience may find easier our information in this special issue, compared to Sensors.

The study is based on a hard surface. What about a soft one? – Thank you for your comment. In order to be used as a screening tool, at the very beginning of a gait analysis exam, that can lead to a more in-depth evaluation depending on the results obtained with the device, the simple straight walk of the horse on a hard surface, at the beginning of the classical lameness exam, was considered by our team to be the most relevant.

Horses are mixed, which may influence the results. – Thank you for your comment. Our intention was to test the device in conditions that are the most similar to those occurring in the every-days of a horse veterinary practice.

As it looks like from figure 3 the devices influence the horse's range of motion. Their positioning is not described in detail in the text. Did you check it anyway? – Thank you for your comment. The attachment system of the devices was made of very flexible straps and also, because the small dimensions and light weight of the devices, this attachment solution exerted only a modest pressure on the limbs, similar for both the lame leg and healthy ones. Moreover, the mobility of the pastern is limited and all horses had been previously accustomed with the wear of protective equipment, namely hoof-boots. Thus, we considered that the influence of the device, if it was one, was similar for all the legs of a given horse and all the assessed horses. This explanation had been included in the manuscript, above Figure 3.

L 267-279 – it is not clear what value is measured and analyzed. Please be as specific as possible. – Thank you for your comment. This aspect has been addressed (at L286 of the manuscript’s corrected version) to provide more clarity for the reader.

L 278 – does it mean that they agree? Not clear. Ethical statement not mentioned here in the text or the last part of the paper. – Thank you for your question. The manuscript’s phrasing on this aspect had been improved for more clarity (at the end of the Materials and methods section).

3. results – table 1 is described, and tables 2-5 are not. What is it m? and others? It should be clear from the tables themselves. – Thank you for your comment. The requested information had been added.
4. discussion – there are many interesting parts, however, the language should be more formal and scientific. Needs revision. – Thank you for your advice. This issue had been addressed.

L 380-382 – interesting part, citations needed. – Thank you for your advice. Citation has been added.

L 409 – please use proper wording (Back and Clayton 2013) for scientific comparison – Thank you for your advice. This issue had been addressed.

L 447 It would be interesting if your data will be also validated by another objective system, not only the subjective judgment of veterinarians. – Thank you for your comment. The reason for the lack of such validation is the same with the main cause for which our device had been constructed: the limitation of material resources available.

5. conclusions – your preliminary results should be concluded. The information that the system will be validated is not the result. – Thank you for your comment. The conclusions had been completed.

Reviewer 2 Report

Lameness detection within the equine sport horse industry is a critical issue and thanks to new technologies there is hope that this area of the industry can be expanded and more effectively addressed with more objective measures. There have been studies as the authors in this manuscript reported that the traditional AAEP scoring method can be subjective and depending on the experience of the veterinarian more subtle lamenesses can be missed. This can be a critical, if not costly problem within the sport horse industry. This is where new technologies can be useful, but the costs, the required technical expertise, and the practicality in the application of these technologies within a clinical setting restrict the potential use of these new technologies outside of the research setting. This is an issue that makes this manuscript very relevant to readers and the technology presented in this manuscript holds promise within the industry. Nevertheless, the flaws within the methodology at this point in the development of this manuscript restrict any conclusions that can be made from what was presented and restricts potential for publication without redoing aspects of the study.

First off, before addressing the methodology issues within this manuscript, it is important to address the activities associated with this study that were "waived" by the institutional review board according to the authors. This was a research study, not a case study performed by veterinarians. Animals were subjected to a device that had not been tested before within a research setting so these were not animals getting veterinary treatment, they were research animals requiring review of procedures and not a waiver of the research protocol. Owners signed off on treatment of animals, not the application of experimental technologies that have never been tested before within a research setting. The authors clearly state in the objectives that this study was to "design, develop, and test an original device system", meaning this is experimental research, not typical treatment practices that have been traditionally utilized within a clinical setting. Veterinarians cannot be using clients’ animals for research without not only consent that is understanding that this is research, not treatment, and without an institutional review board monitoring every aspect of the research protocol. Unfortunately, within the University setting that offers veterinary services this fine line between research and clinical practice can get blurry, but in this case, this clearly falls under the definition of research and not clinical practice. If this was a study based on treatment, thus, a manuscript focused on case studies, then, the manuscript needs to be re-written as a case study where each animal is thoroughly discussed concerning their histories, diagnosis, and treatments. At no point does the reader know what type of health issue created the lameness, where it is located anatomically, and how it was addressed, thus, this is not a case study and requires institutional review board review of the research protocol. Waiver may have been allowed by the board and owner consent may have been given, however, this appears to have been done without thoroughly understanding that experimental procedures were being performed on these animals. While this may be a "specialized university equine clinic" that utilizes new technologies in lameness diagnostics, if the authors are going to present this work as a research study instead of a case study, then, the animals were utilized as research animals. While the authors can go back and rewrite the manuscript as case studies of the animals worked with and discuss how the animals were diagnosed using the new technology, at this point, it is apparent that the work was done more as an experiment, rather than treatment, and thus, draws ethical concerns that would restrict publication utilizing this data. 

Nevertheless, moving past the ethical concerns, the title focuses on "development" of this new technology, but the focus goes well beyond the discussion of development within the manuscript. In general, the manuscript is lengthy and can be more streamlined if the authors stay on track with the title, specifically concerning the "development" aspect. While it is important in the discussion of development to validate that this new technology can be reliable when utilized, this can be more effectively accomplished using one horse, focusing on one specific limb for lameness evaluation. Instead, the authors should utilize a research animal where the lameness is applied after the same subject is tracked without the lameness and where other technologies are applied to be utilized for comparisons such as the force plate and/or video-computer gait analysis. Furthermore, these other technologies should be utilized with the device both on and off the horse for further comparisons. With multiple types of lamenesses on various limbs and with varying degrees of lameness, the variability within this study makes it surprising, if not questionable, that there is any statistical significance.  In addition, with only 10 horses that are of varying breeds and ages, you're looking at additional variability added to the measurements. Finally, with no control on velocity, which has a significant impact on gait, that creates additional concerns as to the reliability of the results. For example, a horse with a more intense lameness could have been less noticeable if it was allowed to track at a slower velocity compared to another horse. Furthermore, how do we know if the device on the limb isn't impacting gait, despite it's smaller size? Was the lameness associated with areas where the device was attached causing further irritation? Using force plates and video-computer gait analysis to not only verify the validity of the device measurements, but to also measure lameness without the device would have been helpful in more clearly validating the reliability of the new technology.

While three "experienced" veterinarians were used for lameness scoring to validate the device, the authors spend a good portion of the manuscript justifying the use of the device by laying out the limitations of visual assessment of lameness, particularly pointing out the issue of subjectivity and the lack of consistency with this method of detecting lameness. In addition, what makes these individuals "experienced" and qualified to validate a new technology? Wouldn't these individuals fall into the same limitations discussed in the introduction concerning consistency and subjectivity within the visual assessment method? In addition, were all three used for every horse and for every pass of the animal where data was collected? Were x-rays and other diagnostic tools utilized to help to validate these scores given by the "experienced" veterinarians? In addition, why was a walk performed as some of the scoring of these animals goes below where a veterinarian would observe consistently lameness at a walk according to the definition of each score given by AAEP? If they did observe consistently lameness at a walk, then, I would argue that according to AAEP definition the horse was not exhibiting a lameness score of 1 or 2, if not even a 3. Also, it is confusing as scores are given for each limb. AAEP scoring is done for the overall appearance of the horse, although there are indicators of where specific areas may be of concern when we perform this scoring, it is through additional diagnostics that we determine the specific location causing the lameness, but at no point are we giving a score specific for each limb of the horse. It's an overall score for the horse, not each limb. This is why comparisons made with proven technologies like a force place or force-measuring shoe would be more appropriate as these systems can measure individually each limb unlike the AAEP lameness scale. Also, why was the lameness scoring done after the use of the new technology? A random design should have been set up as it could be questioned that a lame horse can become worst with further tracking. It's hard to validate measurements if the scoring wasn't done at the same time or done in a random design. It is also a concern that comparisons were made with the "contralateral sound limb" as compensation in the contralateral limb occurs quite frequently, especially within horses exhibiting a higher lameness score and/or ones that have been exhibiting lameness for longer periods. Nevertheless, no history was presented on the animals within this manuscript and tracking specific to each animal was also not presented, and instead, just overall measures for all horses were given. Also, it is important to objectively define what is considered "sound" for this study and how that was objectively determined. 

In addition, as mentioned previously, the new technology needs to be validated using other reliable technologies such as force plates, force-measuring shoes, and/or video-computer based gait analysis. These technologies should be utilized, instead of the AAEP scoring system, measuring both the device on and off the leg to ensure the device does not hinder natural locomotion. It may also be helpful to look at displacement of the device during gait and potential impact of this displacement on measurements, similar to skin displacement issues with tracking markers for video-computer gait analysis, in which the use of these other technologies for comparison purposes could be useful in investigating this concern. Furthermore, as published by research using these other technologies, velocity needs to be controlled during data collection in which a treadmill is the most reliable method, although this limits the use of a force plate. In addition, since each limb was measured, more than five steps needs to be analyzed and this needs to be done with simultaneous analysis performed using the other technologies. If this method is to be eventually utilized in a clinical setting, seeing how the horse potentially adapts to the device will be useful to understand for the clinician in determining standard protocols for use of this device. Nevertheless, if there is a justification for why only five steps were measured, this needs to be clearly addressed. All of these procedures should be added to the research protocol if the objective of the study is to "test" "various lameness degrees", although at this point of the research process the focus should be on the device "design" and "development" with testing coming in a follow up publication.

In general, the new technology presented is of interest to the industry, but care needs to be taken in ensuring that the validation of this method is done in a sound scientific manner so that it isn't discarded by the scientific community. The manuscript needs to be streamlined in length with especially the introduction and discussion shortened and references focused on the most relevant studies. The focus of validation needs to be on one horse with one specific lameness that is created within a research setting so that the horse can be evaluated with the new technology before and after the onset of lameness, thus, a control. Use of just one horse, since the focus is on "development", avoids the introduction of gait influences due to breed type, performance type, and even age-related factors such as arthritic changes within the joints. While the use of one horse will limit the use of statistical analysis, instead of focusing on comparisons between horses and lamenesses, the focus of the manuscript can be on the "development" of the new technology, specifically expanding on the discussion of the device and the computer software as this area is vague within the methodology. While the manuscript lacks a hypothesis statement, which would need to be added for future publication, the objective statement goes beyond the title and is too broad reaching at this point of the development of this new technology. The objective of testing "various lameness degrees" needs to come up later in the research process after the new technology is designed and developed with clear, published, reliable data. Furthermore, keep in mind that during the design and development state of new technology or even a new procedure where research testing is needed for validation, institutional review board approval of all aspects of the protocol is required so that waiving of the procedures should not be an option at this point of the development process. With proper focus on the objective of this stage of the research process, concentrating on the development stage of the new technology, this device can be properly validated and presented to the scientific community so that the usefulness of this new technology won't get discounted because of improper steps within the research methodology. 

Author Response

Lameness detection within the equine sport horse industry is a critical issue and thanks to new technologies there is hope that this area of the industry can be expanded and more effectively addressed with more objective measures. There have been studies as the authors in this manuscript reported that the traditional AAEP scoring method can be subjective and depending on the experience of the veterinarian more subtle lamenesses can be missed. This can be a critical, if not costly problem within the sport horse industry. This is where new technologies can be useful, but the costs, the required technical expertise, and the practicality in the application of these technologies within a clinical setting restrict the potential use of these new technologies outside of the research setting. This is an issue that makes this manuscript very relevant to readers and the technology presented in this manuscript holds promise within the industry. Nevertheless, the flaws within the methodology at this point in the development of this manuscript restrict any conclusions that can be made from what was presented and restricts potential for publication without redoing aspects of the study. -Thank you for your valuable insight.

First off, before addressing the methodology issues within this manuscript, it is important to address the activities associated with this study that were "waived" by the institutional review board according to the authors. This was a research study, not a case study performed by veterinarians. Animals were subjected to a device that had not been tested before within a research setting so these were not animals getting veterinary treatment, they were research animals requiring review of procedures and not a waiver of the research protocol. Owners signed off on treatment of animals, not the application of experimental technologies that have never been tested before within a research setting. The authors clearly state in the objectives that this study was to "design, develop, and test an original device system", meaning this is experimental research, not typical treatment practices that have been traditionally utilized within a clinical setting. Veterinarians cannot be using clients’ animals for research without not only consent that is understanding that this is research, not treatment, and without an institutional review board monitoring every aspect of the research protocol. Unfortunately, within the University setting that offers veterinary services this fine line between research and clinical practice can get blurry, but in this case, this clearly falls under the definition of research and not clinical practice. If this was a study based on treatment, thus, a manuscript focused on case studies, then, the manuscript needs to be re-written as a case study where each animal is thoroughly discussed concerning their histories, diagnosis, and treatments. At no point does the reader know what type of health issue created the lameness, where it is located anatomically, and how it was addressed, thus, this is not a case study and requires institutional review board review of the research protocol. Waiver may have been allowed by the board and owner consent may have been given, however, this appears to have been done without thoroughly understanding that experimental procedures were being performed on these animals. While this may be a "specialized university equine clinic" that utilizes new technologies in lameness diagnostics, if the authors are going to present this work as a research study instead of a case study, then, the animals were utilized as research animals. While the authors can go back and rewrite the manuscript as case studies of the animals worked with and discuss how the animals were diagnosed using the new technology, at this point, it is apparent that the work was done more as an experiment, rather than treatment, and thus, draws ethical concerns that would restrict publication utilizing this data. – Thank you for your valuable comment. The Ethics Committee of the UASVM Cluj-Napoca considered that the procedures described in this manuscript do not fall under the legislation for the protection of animals used for scientific purposes, Romanian national law 43/2014, in accordance with the EUs Directive 2010/63/EU on the protection of animals used for scientific purposes because all the described procedures had been legal veterinary acts (the use of a recording device while the animals had been visually assessed), performed in a clinic. As regards the use of a minimal-invasive gait monitoring and recording device system attached on the legs of the studied horses, taking into account the devices’ design, weight, and shape, and the fact that all horses had been previously accustomed with the wear of feet and leg protections due their sport activity, the Committee considered it under the exemption claim of the abovementioned law (article 1(7), letter f), ‘acts that are not susceptible to provoke pain, suffering, considerable stress or prolonged harm equivalent or above to that produced by the introduction of a needle, in accordance with good veterinary practices’. Moreover, each of the owners have been informed and presented with the use of the device in another horse (owned by the UASVM) and all of them agreed for the non-invasive, about two minutes long procedure to be added at the beginning of the classical lameness assessment procedurally performed at our hospital. To avoid all possible misunderstandings, the owner or her/his delegate was present during the whole lameness evaluation of each horse, including the use of the Lameness detector 0.1. As for the possibility to present this study as a case study, that had not been the intention of the authors.  

Nevertheless, moving past the ethical concerns, the title focuses on "development" of this new technology, but the focus goes well beyond the discussion of development within the manuscript. In general, the manuscript is lengthy and can be more streamlined if the authors stay on track with the title, specifically concerning the "development" aspect. While it is important in the discussion of development to validate that this new technology can be reliable when utilized, this can be more effectively accomplished using one horse, focusing on one specific limb for lameness evaluation. Instead, the authors should utilize a research animal where the lameness is applied after the same subject is tracked without the lameness and where other technologies are applied to be utilized for comparisons such as the force plate and/or video-computer gait analysis. Furthermore, these other technologies should be utilized with the device both on and off the horse for further comparisons. – Thank you for your comment. The main reason to develop a very affordable device to be used as a screening tool at the beginning of the lameness assessment (in order to attract more attention in the following parts of the exam which could be then broadened and made more comprehensive as needed) was the fact that, because material limitations, we did not have access to other means of objective lameness assessment devices and equipment. We do consider that this situation is similar for many clinicians and we consider that they would benefit the most by using this type of device.
With multiple types of lamenesses on various limbs and with varying degrees of lameness, the variability within this study makes it surprising, if not questionable, that there is any statistical significance.  In addition, with only 10 horses that are of varying breeds and ages, you're looking at additional variability added to the measurements. -Thank you for your comment. The group of the horses included in the study was heterogenous, indeed. Our intention was to test the device in conditions that are the most similar to those occurring in the every-days of an equine veterinary practice.
Finally, with no control on velocity, which has a significant impact on gait, that creates additional concerns as to the reliability of the results. For example, a horse with a more intense lameness could have been less noticeable if it was allowed to track at a slower velocity compared to another horse. -Thank you for your comment. All the horses have been led by the same handler, walking by the same length of strides and at the same speed. However, we do agree and understand that this aspect is not enough for standardization of the methodology. On the other hand, the aim of using the Lameness detector 0.1 was not to differentiate between the lameness grades according to the results recorded. The device was meant to only establish if the horse was lame or not, as a screening tool, although the variations in the mean numbers of impulses proved to variate according to the lameness grade of the horses. However, your point of view is very valuable to us and it will be taken into consideration while further improving the device for a more extended use.
Furthermore, how do we know if the device on the limb isn't impacting gait, despite it's smaller size? Was the lameness associated with areas where the device was attached causing further irritation? Using force plates and video-computer gait analysis to not only verify the validity of the device measurements, but to also measure lameness without the device would have been helpful in more clearly validating the reliability of the new technology. -Thank you for your comment. None of the assessed horses had any impairment in the area where the devices have been attached. The attachment system of the devices was made of very flexible straps and also, because the small dimensions and light weight of the devices, this attachment type exerted only a modest pressure on the limbs, similar for both the lame leg and healthy ones. Moreover, the mobility of the pastern is limited and all horses had been previously accustomed with the wear of protective equipment, namely hoof-boots. Thus, we considered that the influence of the device, if it was one, was similar for all four legs of a given horse and in all the assessed horses. This explanation had been included in the manuscript, above Figure 3. As for the use of other objective lameness assessment devices, these have not been available for us.
While three "experienced" veterinarians were used for lameness scoring to validate the device, the authors spend a good portion of the manuscript justifying the use of the device by laying out the limitations of visual assessment of lameness, particularly pointing out the issue of subjectivity and the lack of consistency with this method of detecting lameness. In addition, what makes these individuals "experienced" and qualified to validate a new technology? Wouldn't these individuals fall into the same limitations discussed in the introduction concerning consistency and subjectivity within the visual assessment method? Thank you for your comment. Exactly because the limitations of the subjective lameness assessment methods, individual bias, and lack of consistency, we decided that the visual evaluation made by three experienced veterinarians was a good approach to accept their final mutually agreed lameness score for each horse. As regards the experience and qualification of the assessing veterinarians, they are clinicians who work for at least the past 10 years exclusively in horses, mostly in orthopedics.
In addition, were all three used for every horse and for every pass of the animal where data was collected? Thank you for your question. The answer is yes.
Were x-rays and other diagnostic tools utilized to help to validate these scores given by the "experienced" veterinarians? Thank you for your question. The answer is yes.
In addition, why was a walk performed as some of the scoring of these animals goes below where a veterinarian would observe consistently lameness at a walk according to the definition of each score given by AAEP? If they did observe consistently lameness at a walk, then, I would argue that according to AAEP definition the horse was not exhibiting a lameness score of 1 or 2, if not even a 3. Thank you for your question and comment. The use of the Lameness detector 0.1 was performed at walk each time, and this would be the gait for which we recommend its usage. The rest of the lameness assessment have been performed in the standardized manner recommended by the AAEP.
Also, it is confusing as scores are given for each limb. AAEP scoring is done for the overall appearance of the horse, although there are indicators of where specific areas may be of concern when we perform this scoring, it is through additional diagnostics that we determine the specific location causing the lameness, but at no point are we giving a score specific for each limb of the horse. It's an overall score for the horse, not each limb. This is why comparisons made with proven technologies like a force place or force-measuring shoe would be more appropriate as these systems can measure individually each limb unlike the AAEP lameness scale. Thank you for your comment. The lameness score is given for the animal, as your comment correctly states, but one of our selection criteria for the studied horses (as we described) was for them to be lame in only one leg.
Also, why was the lameness scoring done after the use of the new technology? A random design should have been set up as it could be questioned that a lame horse can become worst with further tracking. It's hard to validate measurements if the scoring wasn't done at the same time or done in a random design. It is also a concern that comparisons were made with the "contralateral sound limb" as compensation in the contralateral limb occurs quite frequently, especially within horses exhibiting a higher lameness score and/or ones that have been exhibiting lameness for longer periods. Thank you for your comment. The lameness scoring with the device have been performed at the beginning of the assessment to avoid the possible aggravation of gait abnormalities due to the effort the horse could have been subjected to, during the assessment. Our reason was to verify if the Lameness detector 0.1 showed the slightest possible lameness.
Nevertheless, no history was presented on the animals within this manuscript and tracking specific to each animal was also not presented, and instead, just overall measures for all horses were given. Thank you for your comment. These aspects were not the aim of our study.
Also, it is important to objectively define what is considered "sound" for this study and how that was objectively determined. Thank you for your comment. “Sound” for this study was the horse with score 0 on the AAEP lameness assessment scale.
In addition, as mentioned previously, the new technology needs to be validated using other reliable technologies such as force plates, force-measuring shoes, and/or video-computer based gait analysis. These technologies should be utilized, instead of the AAEP scoring system, measuring both the device on and off the leg to ensure the device does not hinder natural locomotion. It may also be helpful to look at displacement of the device during gait and potential impact of this displacement on measurements, similar to skin displacement issues with tracking markers for video-computer gait analysis, in which the use of these other technologies for comparison purposes could be useful in investigating this concern. Thank you for your comment. We do understand and agree with your statements, but we did not have access to other objective gait analysis devices.
Furthermore, as published by research using these other technologies, velocity needs to be controlled during data collection in which a treadmill is the most reliable method, although this limits the use of a force plate. In addition, since each limb was measured, more than five steps needs to be analyzed and this needs to be done with simultaneous analysis performed using the other technologies. If this method is to be eventually utilized in a clinical setting, seeing how the horse potentially adapts to the device will be useful to understand for the clinician in determining standard protocols for use of this device. Nevertheless, if there is a justification for why only five steps were measured, this needs to be clearly addressed. Thank you for your comment. We decided to use the device for recording only five steps to ensure a very quick and easy possibility for the clinician in her/his initial screening of lameness of the assessed horse. This information has been stated in the corrected version of the paper.
All of these procedures should be added to the research protocol if the objective of the study is to "test" "various lameness degrees", although at this point of the research process the focus should be on the device "design" and "development" with testing coming in a follow up publication. Thank you for your comment. This issue has been addressed.
In general, the new technology presented is of interest to the industry, but care needs to be taken in ensuring that the validation of this method is done in a sound scientific manner so that it isn't discarded by the scientific community. The manuscript needs to be streamlined in length with especially the introduction and discussion shortened and references focused on the most relevant studies. Thank you for your comment. The introduction and discussions have been shortened.
The focus of validation needs to be on one horse with one specific lameness that is created within a research setting so that the horse can be evaluated with the new technology before and after the onset of lameness, thus, a control. Use of just one horse, since the focus is on "development", avoids the introduction of gait influences due to breed type, performance type, and even age-related factors such as arthritic changes within the joints. While the use of one horse will limit the use of statistical analysis, instead of focusing on comparisons between horses and lamenesses, the focus of the manuscript can be on the "development" of the new technology, specifically expanding on the discussion of the device and the computer software as this area is vague within the methodology. While the manuscript lacks a hypothesis statement, which would need to be added for future publication, the objective statement goes beyond the title and is too broad reaching at this point of the development of this new technology. The objective of testing "various lameness degrees" needs to come up later in the research process after the new technology is designed and developed with clear, published, reliable data. Furthermore, keep in mind that during the design and development state of new technology or even a new procedure where research testing is needed for validation, institutional review board approval of all aspects of the protocol is required so that waiving of the procedures should not be an option at this point of the development process. With proper focus on the objective of this stage of the research process, concentrating on the development stage of the new technology, this device can be properly validated and presented to the scientific community so that the usefulness of this new technology won't get discounted because of improper steps within the research methodology. Thank you for your comment and summarizing the concerned aspects. We do hope we succeeded to answer and/or address them.

Reviewer 3 Report

This is an interesting article for equine orthopedics.

Here my remarks:

The title: "Development of a novel inertial sensor for detection of equine 2 lameness: a preliminary study" is misleading, because the authors used only accelerometers. State of the art IMU's (inertia measurement units) consist of accelerometers, gyroscopes and magnetometers. Please change the title accordingly.

The orientation of the sensors is not clear to me. The authors stated: "The accelerometer was positioned in such way that on its X axis it registered the acceleration of horizontal movement (forward-backward), on its Y axis of vertical movement (upward-downward, free falling), and on its Z axis that of the lateral movement (axial-abaxial or medial-lateral)." 
Did you describe the orientation at the standing horse? During the movement of the limb, the orientation/axis will change in relation to a fixed world coordinate system. Please clarify and add at least coordinate system to extremity/sensor in figure 3 to show the directions (x,y,z) in relation to the standing horse.

Please add a figure where you explain the signal processing systematically (step by step), maybe you can extend figure 4. 
What is meant, if you write about taken impulses (The number of impulses recorded...)? 
Did you mean the sample rate? Please clarify.

Because your sensors are fixed on the proximal phalanx, a rotation in relation to the hoof will occur during the stance phase. Please discuss if an additional gyroscope could be useful to detect lameness.

It would be interesting, if skin displacement and additional movements of the sensors will affect the results. Please discuss the fixation of the sensors.

Author Response

Here my remarks:
The title: "Development of a novel inertial sensor for detection of equine 2 lameness: a preliminary study" is misleading, because the authors used only accelerometers. State of the art IMU's (inertial measurement units) consist of accelerometers, gyroscopes and magnetometers. Please change the title accordingly. – Thank you for your comment. As you correctly made us aware, the IMU’s are constructed using at least accelerometers and gyroscopes together in the unit, both of these being inertial sensors. Exactly for this reason we avoided the IMU term, because referring to the already consecrated notion of ‘unit’ but using only one inertial sensor would not have been correct. Thus, we consider our title accurate.

The orientation of the sensors is not clear to me. The authors stated: "The accelerometer was positioned in such way that on its X axis it registered the acceleration of horizontal movement(forward-backward), on its Y axis of vertical movement (upward-downward, free falling), and on its Z axis that of the lateral movement (axial-abaxial or medial-lateral)." Did you describe the orientation at the standing horse? During the movement of the limb, the orientation/axis will change in relation to a fixed world coordinate system. Please clarify and add at least coordinate system to extremity/sensor in figure 3 to show the directions (x,y,z) in relation to the standing horse. – Thank you for your valuable observation. This issue has been addressed by moving the mentioned paragraph from the accelerometer’s description right after the mentioned figure (Figure 3) and adding more detail to the previously too concise explanation.

Please add a figure where you explain the signal processing systematically (step by step), maybe you can extend figure 4. – Thank you for your comment. The signal processing is explained in the text and we would not like to duplicate that information by transforming it in a visual representation. Also, the textual form may be more understandable by the reader than a schematization of it. Also, additional information has been introduced in the manuscript below Figure 4.

What is meant, if you write about taken impulses (The number of impulses recorded...)? Did you mean the sample rate? Please clarify. – Thank you for your valuable comment. As this aspect may not have been clearly enough explained previously, the issue had been addressed by a short additional description below Figure 4.

Because your sensors are fixed on the proximal phalanx, a rotation in relation to the hoof will occur during the stance phase. Please discuss if an additional gyroscope could be useful to detect lameness. – Thank you for your comment. We had been aware of this aspect but the stance phase (between the hoof-on moment that ends a stride and the hoof-on moment that begins the next stride) did not give us any measurable value as the stance phase is virtually motionless. However, the rotation movement segments have been recorded on the Y axis of the Lameness detector 0.1. With the additional explanations added now on your recommendation about the orientation of the sensors (below Figure 3) this issue had been addressed as well.

It would be interesting, if skin displacement and additional movements of the sensors will affect the results. Please discuss the fixation of the sensors. – Thank you for your comment. We included addition explanation regarding the sensors’ fixation in the manuscript, above Figure 3.

Round 2

Reviewer 2 Report

While the effort and intent of the authors and their research is appreciated, limited revisions were made concerning the concerns of all three reviewers. Although comments from the authors were given to justify reasoning for some of the lack or limited changes associated with the reviewers' comments, each of the reviewers' concerns were legitimate and necessary before proceeding further with this publication. Unfortunately, most of the concerns would require redoing aspects of the data collection in order to made the study more scientifically sound. While the practicality of the study is appreciated as to it's ability to be utilized by clinicians, this does not justify disregarding scientific soundness for publication within a peer-reviewed scientific journal. It may be recommended to look at publication in a more trade-based magazine or website if that is the primary focus of this work. If the authors' aims are to publish within a peer-reviewed scientific journal, the research holds merit and interest if the study is redone utilizing the concerns addressed by all three reviewers and addressing each concern by thorough modification of the data collection procedures.

Reviewer 3 Report

The method is not described comprehensive. Additionally you did not answer my questions, concerning the signal processing and orientation of the sensors.